# Anti-Inflammatory CeO_2_ Nanoparticles Prevented Cytotoxicity Due to Exogenous Nitric Oxide Donors via Induction Rather Than Inhibition of Superoxide/Nitric Oxide in HUVE Cells

**DOI:** 10.3390/molecules26175416

**Published:** 2021-09-06

**Authors:** Mohd Javed Akhtar, Maqusood Ahamed, Hisham Alhadlaq

**Affiliations:** 1King Abdullah Institute for Nanotechnology, King Saud University, Riyadh 11451, Saudi Arabia; mahamed@ksu.edu.sa; 2Department of Physics and Astronomy, College of Sciences, King Saud University, Riyadh 11451, Saudi Arabia; hhadlaq@ksu.edu.sa

**Keywords:** nanoparticles, ROS, superoxide, nitric oxide, signaling effect, endothelial cells

## Abstract

The mechanism behind the cytoprotective potential of cerium oxide nanoparticles (CeO_2_ NPs) against cytotoxic nitric oxide (NO) donors and H_2_O_2_ is still not clear. Synthesized and characterized CeO_2_ NPs significantly ameliorated the lipopolysaccharide (LPS)-induced cytokines IL-1β and TNF-α. The main goal of this study was to determine the capacities of NPs regarding signaling effects that could have occurred due to reactive oxygen species (ROS) and/or NO, since NP-induced ROS/NO did not lead to toxicity in HUVE cells. Concentrations that induced 50% cell death (i.e., IC50s) of two NO donors (DETA-NO; 1250 ± 110 µM and sodium nitroprusside (SNP); 950 ± 89 µM) along with the IC50 of H_2_O_2_ (120 ± 7 µM) were utilized to evaluate cytoprotective potential and its underlying mechanism. We determined total ROS (as a collective marker of hydrogen peroxide, superoxide radical (O_2_^•−^), hydroxyl radical, etc.) by DCFH-DA and used a O_2_^•−^ specific probe DHE to decipher prominent ROS. The findings revealed that signaling effects mediated mainly by O_2_^•−^ and/or NO are responsible for the amelioration of toxicity by CeO_2_ NPs at 100 µg/mL. The unaltered effect on mitochondrial membrane potential (MMP) due to NP exposure and, again, CeO_2_ NPs-mediated recovery in the loss of MMP due to exogenous NO donors and H_2_O_2_ suggested that NP-mediated O_2_^•−^ production might be extra-mitochondrial. Data on activated glutathione reductase (GR) and unaffected glutathione peroxidase (GPx) activities partially explain the mechanism behind the NP-induced gain in GSH and persistent cytoplasmic ROS. The promoted antioxidant capacity due to non-cytotoxic ROS and/or NO production, rather than inhibition, by CeO_2_ NP treatment may allow cells to develop the capacity to tolerate exogenously induced toxicity.

## 1. Introduction

Cerium oxide (CeO_2_) nanoparticles (NPs, particles with at least one dimension less than 100 nm) belong to the rare-earth elements of the lanthanide family. In contrast to many NPs, NPs of CeO_2_ are interesting to researchers due to the antioxidant and anti-inflammatory responses elicited in several in vitro and in vivo models [1,2]. These NPs have exhibited remarkable biocompatibility and protection in human skin fibroblasts against UVA irradiation [3], as well as the protection of cardiac progenitor cells against induced oxidative stress [4] and GSH replenishing activity in human breast MCF-7 cells and fibrosarcoma HT-1080 cells [5]. CeO_2_ NPs have shown excellent cell signaling properties in terms of activating MAPK/ERK, CK2A1, and PKACA-driven survival pathways, leading to the protection of HepG2 cells against damage induced by exogenous H_2_O_2_ [6]. CeO_2_ NPs have recently been shown to be internalized in primary isolated human placental cells [7]. Due to their free-radical scavenging property, CeO_2_ NPs have been implicated in the protection of cells against several kinds of exogenous pro-oxidant agents [2]. Interestingly, these rare-earth NPs have been reported to have mixed effects on angiogenesis [8] and inflammation [9]. Reactive oxygen species (ROS) are reactive forms of oxygen molecules (O_2_). The superoxide radical (O_2_^•−^) and H_2_O_2_ are primary members of ROS that form potent signaling mediators with nitric oxide (NO) at low concentrations while introducing deleterious effects at higher concentrations that exceed the antioxidant capacity of cells [10]. It has been established that ROS could act as signaling mediators due to reversible oxidative modifications of proteins, which can initiate a cascade of phosphorylation reactions leading to altered gene expression [11]. The outcome of such signaling is greatly affected by the type of ROS and its site of production and concentration, as well as by cellular differentiation and types [12,13]. Redox signaling can affect cell survival, proliferation, and programmed cell death [13]. In essence, ROS are dual signaling molecules that result in a diverse array of cell phenotypes ranging from cell survival to death.

In this study, we found that non-toxic concentrations of CeO_2_ NPs induce significant ROS (prominently O_2_^•−^) and NO in human umbilical vein-derived endothelial (HUVE) cells. We hypothesized that the NP-mediated induction of ROS and/or NO could result in the activation of pro-survival signaling, which could prime cells to cope with exogenously induced oxidative stress. Various NADPH oxidases (NOXs, 1–5 and dual oxidases (DUOXs 1-2)) are found in the cell membrane, endoplasmic reticulum, mitochondrial membrane, and nuclear membrane, leading to the accumulation of different ROS in different compartments and resulting in different endpoint-effects or outcomes [14]. Endothelial NOXs and DUOXs have been well documented to participate in cell proliferation signaling by generating O_2_^•−^, causing the activation of tyrosine kinases and protein phosphorylation [15]. We therefore evaluated total ROS (as a collective marker of H_2_O_2_, O_2_^•−^, and hydroxyl radical) by DCFH-DA. An O_2_^•−^ specific probe DHE was employed to decipher the prominently acting members of ROS. NO was also evaluated in the response to NP treatment. To further strengthen the understanding of the aforementioned objectives, two NO donors were applied in human umbilical vein-derived endothelial (HUVE) cells in conjunction with CeO_2_ NPs. To the best of our knowledge, there is limited modulatory potential of CeO_2_ NPs against NO. The two NO donors used in this study were SNP (sodium nitroprusside, #ab145732 from Abcam plc, Cambridge, UK) and DETA-NO (Z)-1-[N-(2-Aminoethyl)-N-(2-ammonioethyl)amino]diazen-1-ium-1,2-diolate or DETA-NONOate #ab144627 from Abcam plc, Cambridge, UK). Human umbilical vein endothelial (HUVE) cell lines were used in this study since endothelial cells that line the circulatory vasculature depend on ROS as well as NO production for diverse physiological functions such as angiogenesis and endocytosis. To achieve the above-mentioned objectives, this study began by freshly synthesizing and characterizing the NPs of CeO_2_.

## 2. Results and Discussion

### 2.1. Physico-Chemical Characterization of CeO_2_ NPs

The average diameter of NPs was calculated by measuring over 100 particles in random fields of TEM view. The size of CeO_2_ NPs was determined to be 37 ± 9 nm when observed by TEM. The TEM image is presented at a resolution of 50 nm (Figure 1A). A matte texture was evidenced in high-resolution TEM (HR-TEM) images captured at 2 nm (Figure 1B), clearly confirming the typical planes found in crystals. The HRTEM image (Figure 1B) of CeO2 NPs suggests a crystalline nature with an interplanar spacing (d) of around 0.255 nm, corresponding to a lattice plane of (111). CeO_2_ NPs appear to be heterogenous in size, but the shape is rhomboid or cubical overall in TEM images, which is in close agreement with SEM images (Figure 1C). Elemental dispersive spectrum (EDS) analysis (Figure 1D) confirms the chemical composition of CeO_2_ NPs, with no impurities detectable. The dynamic light scattering system suggested a particle distribution of 138 ± 67 nm and zeta potential of −24 ± 2 eV in complete culture media, whereas the same measures were 290 ± 104 nm and −11 ± 1.3 eV in sterile distilled water, respectively. Particle agglomeration was significantly lower in complete culture media than in distilled water, suggesting a significant interaction of NPs with culture components, leading to a fair suspension therein. A variety of culture components, especially serum proteins, are known to react with NP surfaces, leading to changes in the primary surface characteristics of naïve NPs to secondary surface characteristics of NPs suspended in the relevant aqueous media [16]. Apart from surface modification by culture components, some inherent properties of NPs, such as their redox activity, play a critical role in bio response. Note that NPs of CeO_2_ are redox-active because of the oscillation in their redox status (from 3^+^ to 4^+^) exhibited by the active component (Cerium) in the NP [17]. Lattice defects that result from Ce^3+^ ions and the compensation by oxygen vacancies on NP surfaces cause redox switching between CeO_2_ and CeO_2-x_ in redox reactions, making these NPs highly catalytic [17]. Unlike non-redox active NPs, variations in the physicochemical properties of CeO_2_ NPs arising due to different synthesis procedures can significantly alter their biological response from antioxidant to pro-oxidant [17,18].

### 2.2. CeO_2_ NPs Did Not Exhibit Toxicity in HUVE Cell but Differentially Protected Cells against the IC50 of Two NO-Donors

NPs of CeO_2_ did not cause cytotoxicity in HUVE cells when exposed for 24 h at concentrations of 50, 100, 200, and 400 µg/mL (Figure 2A). The same was observed for 48 h exposure (data not shown). Then, we evaluated IC50s for DETA-NO and SNP in HUVE cells for 24 h of exposure and examined the protective potential of CeO_2_ NPs, if any was shown, at the concentration of 100 µg/mL. This concentration was chosen since this was the concentration that appeared to be most dynamic in terms of safety and was the lowest concentration responsible for the induction of other biomarkers such as O_2_^•−^ or NO. In addition to MTT, we present calceinAM images of cell viability (green fluorescence images in Figure 2C) and phase-contrast images (black–white images in Figure 2C). CeO_2_ NPs caused a significant recovery in cell viability of up to 77% when determined by MTT and 76% when determined with calceinAM against the IC50 of DETA-NO, whereas the recovery was 92% when determined by MTT and 87% when determined with calceinAM against the IC50 of SNP. CeO_2_ NP improved cell viability to 89% when measured by MTT and 93% when determined by calceinAM against the IC50 treatment of H_2_O_2_. CeO_2_ NP-treated HUVE cells were significantly protected from the toxicity caused by the IC50s of the two NO donors. The data suggest that CeO_2_ NPs exhibited low recovery potential against the prolonged release of the NO donor DETA-NO, whereas it showed higher recovery potential against SNP.

### 2.3. CeO_2_ NPs Significantly Reduced Inflammatory Markers in HUVE Cells

CeO_2_ NP treatment did not result in significant cytokine production in comparison with control cells (Figure 3). Although NPs did not completely prevent the LPS-induced inflammatory markers, the reduction in inflammation induced by LPS was significant in HUVE cells (Figure 3). NPs caused an inhibition of IL-1β by 12% (Figure 3B) and TNF-α by 21% (Figure 3C). We did not find pro-inflammatory activity of CeO_2_ NPs in HUVE cells. As expected, LPS caused significant activity of both IL-1β and TNF-α when exposed for 24 h. Under normal physiological levels, depending on the cell types and other conditions, NO along with O_2_^•−^ can exert potent anti-inflammatory effects, but NO can also be strongly pro-inflammatory under the condition of its over-production [19]. NO-donating drugs acting as NOS inhibitors have been reported to cause the release of NO, inhibit IL-1β, and protect against apoptosis in human endothelial and monocytes [20]. CeO_2_ NPs seem to protect against the cytotoxicity induced by NO-donating and ROS-releasing agents by activating pro-survival signaling through the NP-mediated release of ROS/NO. CeO_2_ NPs have been reported to inhibit the activity of NF-kB, inflammatory cytokines IL-1β and IL-6 production, and NO in these cells that otherwise occurred due to cigarette smoke extract exposure [21]. Similarly, CeO_2_ NPs efficiently protected against an exogenous insecticide, Fipronil, administered in rat, and caused the down-regulation of Fipronil-induced lipid peroxidation, ROS, caspase 3 activity, IL-1β, and NOS activity [22].

### 2.4. CeO_2_ NPs Increased Intracellular O_2_^•−^ in HUVE Cells

Interestingly, CeO_2_ NPs caused no significant cytotoxicity at up to 400 µg/mL when treated for 24 h (and 48 h) but caused a significant induction of ROS at concentrations of 100–400 µg/mL in HUVE cells when exposed for 24 h (Figure 4). We measured ROS by DCF-DA, which non-specifically measures several species of ROS (Figure 4A) and DHE and specifically detects O_2_^•−^ (Figure 4B). In this way, the DCF probe suggested the 1.17-fold, 1.2-fold, and 1.21-fold induction of ROS by CeO_2_ NPs at concentrations of 100, 200, and 400 µg/mL, respectively, without any significant induction at 50 µg/mL of NPs (Figure 4A). When ROS was measured by the DHE probe, its induction started significantly at concentrations from 50 µg/mL, at which the rate of increase in induction was observed to be 1.26-fold, while a 100 µg/mL concentration induced a 1.3-fold increase (Figure 4B). No significant increase in DHE fluorescence was observed at the concentrations of 200 and 400 µg/mL from that caused by 100 µg/mL of NPs. In contrast to DHE fluorescence, DCF intensity appeared in a time-dependent manner only at 24 and 48 h of exposure to 100 µg/mL of NPs, with no effect at 6 and 12 h of exposure (Appendix A). However, induction of O_2_^•−^ was shown much earlier and was more pronounced than for the other ROS (S1-B). The data suggest that the O_2_^•−^ production mediated by non-cytotoxic CeO_2_ NP was significantly lower than that caused by the IC50s of the two NO donors and H_2_O_2_, as evidenced by the DHE measurement (Figure 4C,D). Moreover, the enlarged images in Figure 4C suggest an extranuclear, dim but significant fluorescence of DHE in the cytoplasm of NP-treated cells, which is missing in control cells and all other treatments lacking NP exposure. This dim fluorescence is present in cells treated with NPs alone as well as with agents of NO donors and H_2_O_2_ if combined with NPs, indicating dim fluorescence (scattered over the cytoplasm of cells), which is an NP-dependent characteristic. The same image of the DHE, replacing enlarged images with superimposable blue fluorescence, is provided as S2 (Appendix A) for additional support and clarity. O_2_^•−^ generating CeO_2_ NPs caused a reduction of O_2_^•−^ production due to the IC50s of the two NO donors and H_2_O_2_, thus suggesting a possible mechanism and implications for the NP-mediated production of O_2_^•−^, which are significantly different from those caused by exogenous factors. Such contradictory behavior of inducing oxidative stress but not translating this into corresponding cytotoxicity by CeO_2_ NPs has been reported by several investigators. A permanent increase in ROS at higher concentrations (400 µg/mL) of CeO_2_ NPs in human lens epithelial cells (HLECs) exhibited classic hallmarks of apoptosis, whereas the same NPs protected cells against induced oxidative stress at low concentrations (100 µg/mL) [23]. CeO_2_ NPs have been reported to significantly increase ROS and lipid peroxidation without having adverse effects on cell viability and morphology in HaCaT and A549 cells up to the concentration of 200 µg/mL [24]. In fact, investigators found CeO_2_ NPs to induce oxidative stress at higher concentrations (≤200 µg/mL) but decrease oxidative stress at lower concentrations (≥100 µg/mL) without affecting cell viability in either condition for a 24 h exposure duration [24]. In the present study, a similar trend of an increase in ROS and RNS generation was found, but no sign of significant toxicity was observed up to a concentration of 400 µg/mL when exposed for 24 h in HUVE cells. We believe at the moment that O_2_^•−^ production, in conjunction with NO (discussed later), could increase cells’ antioxidant status from the basal level increasing tolerance against the toxicity of NO donors.

### 2.5. CeO_2_ NPs Increased Low Concentrations of Intracellular Nitric Oxide

NO production due to CeO_2_ NPs and other exposures was detected under DAR-2-labelled live-cell imaging (Figure 5A,B) and by the indirect method of NO determination by quantifying nitrite by Griess reagent (Figure 5C). SNP-mediated NO in HUVE cells was found to be most pronounced, as indicated in Figure 5; it was increased 1.71-fold by DAR-2 imaging and 1.46-fold by Griess reagent. In lieu of end-point effects, CeO_2_ NPs seem to work as moderators in the production of NO, as it efficiently (not completely) ameliorated NO production by NO donors, while simultaneously producing a significant amount of NO when present alone. It was increased 1.33-fold by DAR-2 imaging and 1.24-fold by the Griess reagent. It should be noted that DETA-NO treatment led to a higher measurement of NO production by the Griess reagent (1.34-fold compared to the control) in comparison with DAR-2 imaging (1.23-fold compared to the control). NO is known to exert opposing effects in cell signaling output in terms of life and death either by turning on programmed death pathways or by shutting them off [25]. The data obtained in this study clearly suggest that CeO_2_ NPs are a modulator of NO production in HUVE cells; thus, these NPs greatly affect NO-dependent physiological functions such as endocytosis and angiogenesis that are frequently carried out in various endothelial cells [8]. NO was found to be induced equally by DETA-NO and H_2_O_2_ when measured by DAR-2 imaging, at rates much lower than that induced by SNP. The level of nitrite induced by DETA-NO was higher than that induced by H_2_O_2_ when measured by the Griess reagent, but it was still lower than that induced by SNP. CeO_2_ NP treatment has been reported to induce NO generation via the Griess method in human neuroblastoma (IMR32) cells due to concentrations of 100 and 200 µg/mL of CeO_2_ NPs with 24 h of exposure [26]. However, microparticles of CeO_2_ did not exhibit NO induction phenomenon in the same cell type [26]. Some investigators have found increased NO in the lung and spleen of mice but a decreased NO in the heart [27]. It is clear from several studies that O_2_^•−^ could be a major ROS that can prime cells via autophagic survival against the harmful impacts of ROS [28]. It is also worth noting that intracellular O_2_^•−^ can be converted to another signaling molecule, H_2_O_2_, by superoxide dismutases. Alternatively, the relatively low induction of even mitochondrial O_2_^•−^ may meet signaling requirements via the mitochondrial conversion and emission of H_2_O_2_ [29]. Taken together, the findings on NO and ROS clearly suggest that CeO_2_ NPs result in O_2_^•−^ and NO production in cellular sites, possibly in a way that could lead to a survival signaling rather than toxicity.

### 2.6. Loss of Mitochondrial Membrane Potential (MMP) That Occurred Due to NO Donors Was Significantly Recovered by CeO_2_ NPs

The data clearly suggest that CeO_2_ NPs treatment causes a gain in MMP in comparison with control cells (Figure 6A,B). This trend is also reflected by the finding that the loss in MMP caused by two NO donors and H_2_O_2_ significantly increased in the presence of CeO_2_ NPs. The data further suggest that CeO_2_ NP treatment significantly increased the MMP loss caused by SNP and H_2_O_2_ treatments. The data in this study clearly suggested that CeO_2_ NP treatment had no harmful effect on MMP in comparison with control cells, but rather a gain in MMP was observed due to NPs exposure. This trend is also reflected by the finding that the loss in MMP caused by the two NO donors and H_2_O_2_ significantly increased in the presence of CeO_2_ NPs. A previous study showed an approximately 2- to 3-fold increase in DHE fluorescence (i.e., levels of O_2_^•−^) in NAC-treated mouse 3T3 fibroblasts in comparison with the control followed by a late effect of an increase in Mn superoxide dismutase (SOD2) activity [30]. More importantly, it is clear that the NP-induced increase in O_2_^•−^ production did not have a damaging impact on MMP in HUVE cells, suggesting that NP-mediated O_2_^•−^generation is extra-mitochondrial. NO donors and H_2_O_2_ treatment caused a significant increase in O_2_^•−^ production as well as a significant loss in MMP, suggesting mitochondrial O_2_^•−^ generation. The cell plasma membrane is a natural source of ROS signaling that harbors the elements of ROS production (e.g., NADPH oxidases), a plethora of receptors and signaling proteins and molecules that participate in integration and transmission down the signaling stream [31]. As is quite clear from the enlarged images of DHE fluorescence, O_2_^•−^ generation was found over the cytoplasm of cells treated with NPs alone or in combination. This fluorescence was unique to NP-treated cells only and not present in control cells or cells not receiving CeO_2_ NPs. In essence, the beneficial and detrimental effects of ROS appear to be complex phenomena that depend on biological species and the cellular environment [32].

### 2.7. CeO_2_ NPs Significantly Restored Depleted GSH in Cells and Increased the Activity of Antioxidant Enzymes

CeO_2_ NP treatment caused an increase in GSH concentration in HUVE cells (Figure 7A), mostly at 100 µg/mL of NP, above which this tendency was not particularly pronounced and returned back to control level at 400 µg/mL of CeO_2_ NPs. Although both NO donors at their IC50s caused a significant depletion of GSH, the SNP-mediated GSH depletion was much steeper than that caused by DETA-NO (Figure 7B). The activities of the antioxidant enzyme GR were significantly elevated in cells exposed to 100 µg/mL of CeO_2_ NPs for 24 h (Figure 7C). It is important to remember that GR is an important enzyme for the recycling of GSH from its oxidized form, GS-SG. The activity of GPx was significantly reduced in exogenous toxicant-treated cells (Figure 7D). However, the activity of GPx seemed to be unaltered due to NP (Figure 7D) but was restored due to NP in toxicant-treated cells up to the level present in control cells. Moreover, the data on GR and GPx activities partially explain the mechanism behind the NP-induced gain in GSH as well as persistent cytoplasmic ROS as measured by DHE fluorescence. Investigators have demonstrated an increase in cellular GSH by an increase in nanomolar cellular concentrations of NO [33]. The GSH level was elevated by increasing the amount of NO that occurred either due to the overexpression of NOS or exogenous application of NO donors [34]. Additionally, studies have reported the prevention of exogenous H_2_O_2_-mediated toxicity in many endothelial cells via GSH induction due to low NO signaling [35]. The combination of activated GR and unaltered GPx in the present report is well supported by the high level of GSH replenishment and persistent ROS due to non-toxic NPs. 

## 3. Materials and Methods

Fetal bovine serum, penicillin–streptomycin, and LTR (LysoTracker™ Red DND-99) were purchased from Invitrogen Co. (Carlsbad, CA, USA). DMEM F-12, MTT [3-(4,5-dimethyl thiazol-2-yl)-2,5-diphenyl tetrazolium bromide], NADH, pyruvic acid, perchloric acid, DHE, DCFH-DA, MDC (monodansylcadaverine), Rh123, GSH, o-phthalaldehyde (OPT), substrates for caspases, Hank’s balanced salt solution (HBSS), and Bradford reagent were obtained from Sigma–Aldrich, MO, USA. DAR-2 and kits for IL-1β and TNF-α were purchased from Abcam. Ultrapure water was taken from a Milli-Q system (Millipore, Bedford, MA, USA). All other chemicals used were of reagent grade.

### 3.1. Synthesis and Characterization of CeO_2_ Nanoparticles

CeO_2_ nanoparticles were synthesized by a slightly modified method as described elsewhere [36]. NP synthesis was started by mixing hydrated cerium chloride and sodium carbonate in the molar ratio of 2:3 and placed in a nylon vial. Then, the mixture was subjected to milling at room temperature using a ball-to-powder mass ratio of 5:1. The milled powder was calcined at 700 °C. NaCl was washed with hot deionized water and ethanol successively, using an ultrasonic bath and a microporous press filter. The washed powder was dried in an oven at 75 °C for 3 h and then subjected to analysis. Sizes and shapes were evaluated by transmission electron microscopy (TEM) and scanning electron microscopy (SEM). The chemical composition of NPs was confirmed by performing energy dispersive spectroscopy (EDS) analysis. Dynamic Light Scattering (DLS) measurement was taken in a Nano-Zeta Sizer-HT (Malvern Instruments, Malvern, UK) with freshly suspended CeO_2_ NPs in complete culture media and distilled water at a concentration of 100 µg/mL as most biological studies were carried out at this concentration.

### 3.2. Cell Culture and Cell Viability Due to CeO_2_ NP and NO-Donors

HUVE cells (ATCC, US) were cultured in DMEM supplemented with additional endothelial growth components (CADMEC, Cell Applications, Inc., San Diego, CA, USA). The cells were passaged every 3–4 days before reaching 80% confluency. Cells were exposed with either CeO_2_ NPs or with exogenous ROS/NO donors for a period of 24 h. Cell viability was determined by an MTT [37] assay in which the absorbance of centrifuged supernatant containing solubilized formazan was measured at 570 nm in a plate reader (Synergy HT, Bio-Tek, Winooski, VT, USA). Cell viability was calculated as a percentage of the control. IC50 calculations were performed with the online IC50 calculator (https://www.aatbio.com/tools/ic50-calculator, 12 March 2021) provided by AAT Bioquest, Inc. (CA 94085, USA). Cells were also imaged by phase contrast microscopy.

N.B. A cytotoxic concentration of cell permeable NO-donors, which induced cytotoxicity in HUVE cells by 50% (i.e., IC50s) was evaluated for a 24 h of exposure. It is worth noting that toxic concentrations of DETA-NO and SNP are known to vary with types of cells. For example, the IC50 of DETA-NO varies from 100 µM [38] to 1000 µM [39]. Similarly, IC50 of SNP varies from 500 µM in RAW 264.7 cells [40] to 1000 µM in PC12 cells [41]. In the present study, the IC50s of two NO donors (DETA-NO- 1250 ± 110 µM and SNP- 950 ± 89 µM) were utilized to evaluate CeO_2_ NPs’ cytoprotective potential and the underlying mechanism in HUVE cells. The exogenous IC50 of H_2_O_2_ (120 ± 7 µM) was applied in every experiment as a positive control of induced oxidative stress.

### 3.3. Determination of Cytokines IL-1β and TNF-α Levels

HUVE cells were grown in 24-well plates and treated as described above. After 24 h of exposure of modulators, cell culture supernatant was collected by centrifugation at 3000 rpm for 5 min at a room temperature of 24 °C. In the collected supernatant, IL-1β and TNF-α were examined according to the protocols of the ELISA kits (ab181421, human TNF-α ELISA kit; ab100562, human IL-1β ELISA kit; ab139484, autophagy assay kit) from the manufacturer (Abcam, Discovery Drive, Cambridge Biomedical Campus, Cambridge, CB2 0AX, UK). A specific antibody was employed in the measurement with the IL-1β ELISA kit (#ab100562). Cell culture supernatants (100 µL) from control and treated cells were pipetted into the antibody-immobilized insert-wells and incubated for 3 h at room temperature. After washing wells four times with the supplied wash solution, 100 µL of biotinylated anti-human IL-1β antibody was added for 1 h with gentle shaking. After washing away unbound biotinylated antibody with wash solution four times, HRP-conjugated streptavidin was pipetted to the wells and left for 45 min. After washing again four times, a TMB substrate solution (100 µL) was added for 30 min. Absorbance at 450 nm was immediately taken after adding stop solution (50 µL). A standard of IL-1β was prepared in assay diluent B and run similarly for calculation purposes. 

### 3.4. Determination of Intracellular ROS

The potential induction of ROS was determined by 2’,7′-dichlorofluorescin diacetate (DCFH-DA) probe [42] that was incubated for 45 min at the final concentration of 50 µM after the treatment period was over. The plate was washed thrice with cold HBSS to remove excess dye from each well, and the intensity of DCF fluorescence was taken at 528 nm in the plate reader (Synergy HT, Bio-Tek, Winooski, VT, USA). Dihydroethidium (DHE), a cell permeable probe, was utilized to measure O_2_^•−^, producing the red fluorescent products ethidium and/or 2-hydroxyethidium [43]. Cells were labelled with DHE at a final concentration of 5 μM and incubated for 30 min. Plates were carefully washed with cold HBSS once as this probe had a very high signal-to-noise ratio and did not produce any auto-fluorescence if present outside cells. Fluorescence readings were imaged in a fluorescence microscope using suitable filters (Leica DMi8, Wetzlar, Germany). 

### 3.5. Analysis of Intracellular NO

Intracellular NO was determined by imaging a rhodamine-based live cell-permeable fluorescent probe DAR-2 that reacted specifically with NO and generated intense fluorescence in the infra-red region [44,45,46]. Cells were treated with respective agents for 24 h in a 12-well plate and labeled with DAR-2 at a final concentration of 10 μM for 2 h at the end of treatment. Then, cells were carefully washed with cold HBSS three times, and imaging was conducted using an appropriate filter in a microscope (Leica DMi8, Wetzlar, Germany). The direct imaging of NO using DAR-2 probes was successfully reported in RAW264.7 cells [47], PC-12 cells [48], and Zebrafish [49]. NO was also indirectly quantified by measuring nitrite liberated in cell culture media using the Griess reagent at 540 nm in a plate reader (Synergy HT, Bio-Tek, Winooski, VT, USA). A standard of sodium nitrite (1–100 µM) prepared in culture media was similarly run for calculation purposes as conducted by various investigators [50,51]. Data are presented as the percentage of NO concentration in untreated control cells.

### 3.6. Determination of Mitochondrial Membrane Potential by Rh123

Rhodamine (Rh) 123 is a powerful probe for monitoring the abundance and activity of mitochondria [52,53]. To conduct this assay, Rh123 at a final concentration of 5 μM was added to the cells in 12-well plates for 20 min. The reaction mixture was removed, and cells were washed with HBSS three times and placed for 2 min on a shaker for careful, gentle shaking so that cells would not be washed off. Imaging was conducted under a blue light exciting filter in a microscope (Leica DMi8, Wetzlar, Germany). The resultant green fluorescence intensity was directly proportional to the MMP; the higher the green Rh123 fluorescence, the greater the MMP, and vice-versa.

### 3.7. Determination of GSH and GSH-Related Antioxidant Enzymes

The cellular content of GSH was quantified according to the method given by Hissin and Hilf [54]. After treatment, cells were lysed in an aqueous solution of 0.1% deoxycholic acid plus 0.1% sucrose for 2 h, subjected to 3 freeze–thaw cycles, and centrifuged at 10,000× *g* for 10 min at 4 °C. Supernatant was precipitated in the final concentration of 1% perchloric acid and centrifuged at 10,000× *g* for 5 min at 4 °C. Twenty microliters of the perchloric acid protein-precipitated cell lysate supernatant was mixed with 160 μL of 0.1 M K-phosphate–5 mM EDTA buffer, pH 8.3, and 20 μL o-phthalaldehyde (OPT, 1 mg/mL in methanol) in black 96-well plates. After 2 h of incubation at room temperature in the dark, fluorescence was measured at an emission wavelength of 460 nm (Synergy HT, Bio-Tek, Winooski, VT, USA). A standard curve was obtained for calculation from a similarly prepared known concentrations of GSH. The protein in the cell samples was estimated from unprecipitated supernatant, and data were converted to GSH with the nmol/mg protein present in unprecipitated supernatant.

Cell lysate was prepared to assay glutathione reductase (GR) and glutathione peroxidase GPx by four freeze–thaw cycles in distilled water containing deoxycholic acid and sucrose. Finally, lysate was centrifuged at 14,006× *g* for 10 min at 4 °C. The supernatant was transferred to another tube and protein content was measured. From this cell lysate, the activities of glutathione reductase (GR) and glutathione peroxidase (GPx) were determined as briefly described below. The activity of GR was determined by the method of monitoring the oxidation of NADPH in its reaction with oxidized glutathione (GS-SG) in the reaction buffer (0.2 M KH_2_PO_4_ buffer, pH 7.3, 2 mM GS-SG, 0.3 mM NADPH) as described elsewhere [55]. GR activity was expressed as nmol NADPH/min/mg protein. The activity of GPx was determined by the method described elsewhere [56] in which the reducing potential of GSH was coupled in the reduction of cumene hydroperoxide. A continuous supply of GSH was ensured by adding GR enzyme and NADPH in the reaction buffer (0.1 MTris–HC1 buffer, pH 8.0, 1 mM EDTA, 3 mM GSH, 0.2 mM NADPH, 0.5 units of glutathione reductase/mL). GPx activity was expressed as a function of NADPH oxidation as nmol NADPH/min/mg protein.

### 3.8. Protein Estimation

The total protein content was measured by using a convenient BCA Protein Assay Kit from Sigma-Aldrich as per the instructions.

### 3.9. Statistics

An ANOVA (one-way analysis of variance) followed by Dunnett’s multiple comparison tests was employed for the statistical analysis of results. For a particular set of the experiment, a burst of images was captured at constant exposure rates in terms of time, gain, saturation, and gamma. For the calculation of corrected total cellular fluorescence (CTCF), a reasonably constant area was selected and restored via the “restore selection” command for all images once opened in ImageJ software (NIH, Bethesda, MD, USA). CTCF was calculated by subtracting fluorescence in the background (without cell) from the mean of individual cellular fluorescence. The scale bar in images was set using ImageJ after adjusting the scale of pixels/microns for a particular objective and then saving all images in a JPEG format. Representative images were captured with a 5-megapixel Leica DFC450C camera (Wetzlar, Germany).

## 4. Conclusions

Our study suggests that if O_2_^•−^ production occurs due to cytoprotective CeO_2_ NPs, it might be extra-mitochondrial. This kind of O_2_^•−^ production, in conjunction with NO, could increase cells’ antioxidant status from the basal level. This promoted antioxidant capacity, which, as evidenced by an increase in GSH levels in NP-treated cells, is involved in allowing cells to tolerate the induced toxicity that might be due to exogenous oxidants and other toxic insults. In essence, CeO_2_ NPs’ cytoprotective pathways seem to be activated via generating, rather than inhibiting, ROS and NO. CeO_2_ NPs are not cytotoxic but are sufficient to prime cells to tolerate toxicants to varying degrees. This study supports the further investigation of CeO_2_ NPs synthesized by different methods to determine their potential to modulate pro- and anti-survival pathways in order to confirm the mechanism of the biological response of this promising NP.

## Figures and Tables

**Figure 1 molecules-26-05416-f001:**
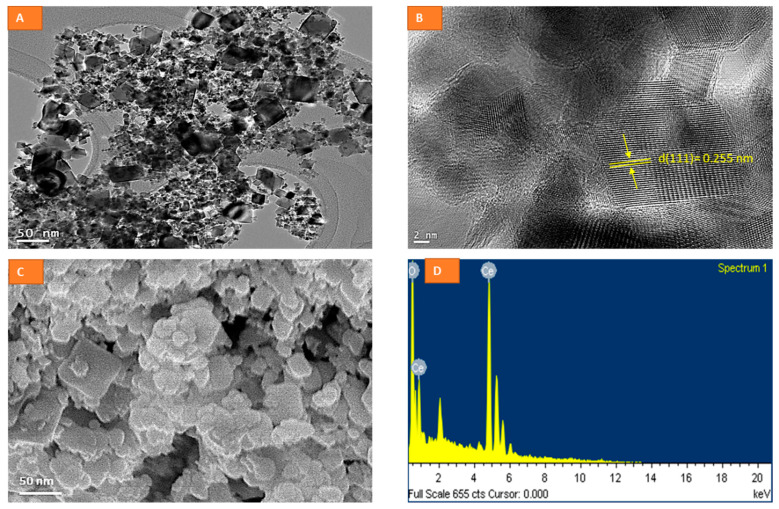
Sizes of CeO_2_ NPs characterized by TEM (**A**), crystal planes by TEM taken at a 2 nm resolution (**B**), shape by SEM (**C**), and composition by EDS (**D**).

**Figure 2 molecules-26-05416-f002:**
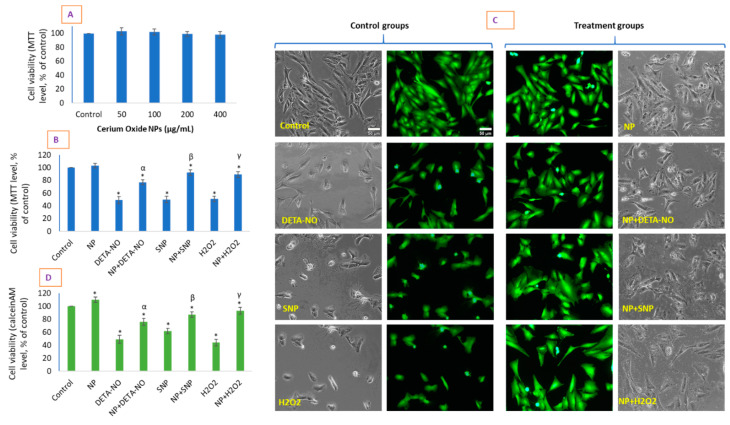
Cell viability determined by MTT due to 24 h exposure of CeO_2_ NPs in a concentration-dependent manner (**A**). Concentrations are given in µg/mL. HUVE cells were treated with the carefully determined IC50s of two NO donors: DETA-NO and SNP. H_2_O_2_ at its IC50 for 24 h was used as a positive control of ROS. The cytoprotective potential, if any, of CeO_2_ NPs (chosen at 100 µg/mL of CeO_2_ NPs) against the IC50 of the above exogenous RNS/ROS was evaluated again by MTT (**B**). (**C**) represents live cell images labelled with a calceinAM probe (green images) with corresponding phase-contrast images. The intensity of calceinAM fluorescence is given in (**D**) as calculated in the ImageJ software. The scale bar, marked only in control images, represents 50 µm captured by a 20× objective. Data represented are means ± SD of three identical experiments (*n* = 3) made in triplicate. * Statistically significant difference as compared to the controls (*p* < 0.05). α, β, and γ, if present, denote a significant cytotoxicity recovery response in the presence of NP (as 100 µg/mL of CeO_2_ NPs) against 50% cytotoxicity caused by (IC50s of) DETA-NO, SNP, and H_2_O_2_, respectively (*p* < 0.05).

**Figure 3 molecules-26-05416-f003:**
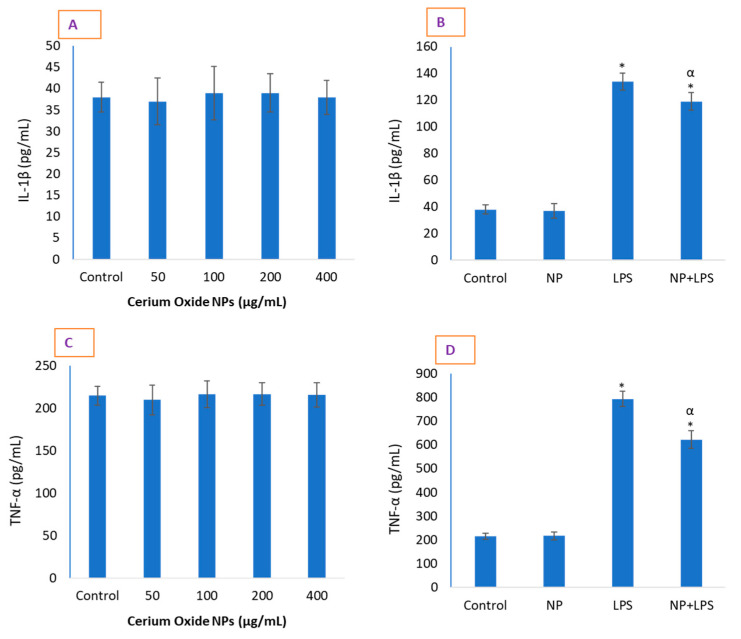
Data suggest that CeO_2_ NPs do not induce inflammatory marker IL-1β (**A**). Conversely, CeO_2_ NPs significantly ameliorated LPS (lipopolysaccharide)-induced IL-1β (**B**). A similar phenomenon is shown for TNF-α (**C**,**D**). It should be noted that the concentration of LPS used was only inflammatory, not cytotoxic. CeO_2_ NP co-treatment, however, did not cause a complete reduction in these markers, which were elevated due to LPS exposure. Data represented are means ± SD of three identical experiments (*n* = 3) made in triplicate. * Statistically significant difference as compared to controls (*p* < 0.05). α denotes a significant anti-inflammatory potential of NPs against LPS in term of either IL-1β or TNF-α (*p* < 0.05).

**Figure 4 molecules-26-05416-f004:**
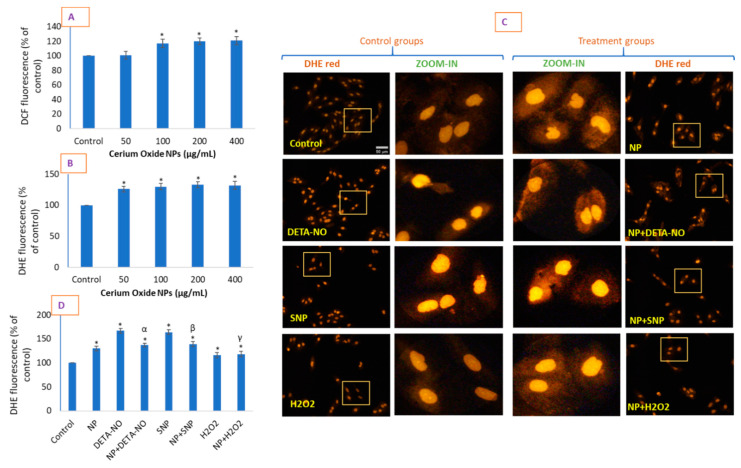
Production of ROS was evaluated by DCFH-DA (**A**) and DHE (**B**) probes, which are known to react non-specifically with several species of ROS and O_2_^•−^, respectively. The magnitude of DHE fluorescence was significantly higher than that obtained by DCF fluorescence. Cells further treated with exogenous RNS/ROS and were live imaged under DHE fluorescence (**C**), and the fluorescence intensities from individual cells in each cell group were quantified using image J software (**D**). Yellow squares represent corresponding enlarged areas. The scale bar, marked only in control images, represents 50 µm captured by a 20× objective. Data represented are means ± SD of three identical experiments (*n* = 3) made in triplicate. * Statistically significant difference as compared to controls (*p* < 0.05). α, β, and γ, if present, denotes a significant NP-mediated inhibition of ROS caused by DETA-NO, SNP, and H_2_O_2_, respectively (*p* < 0.05).

**Figure 5 molecules-26-05416-f005:**
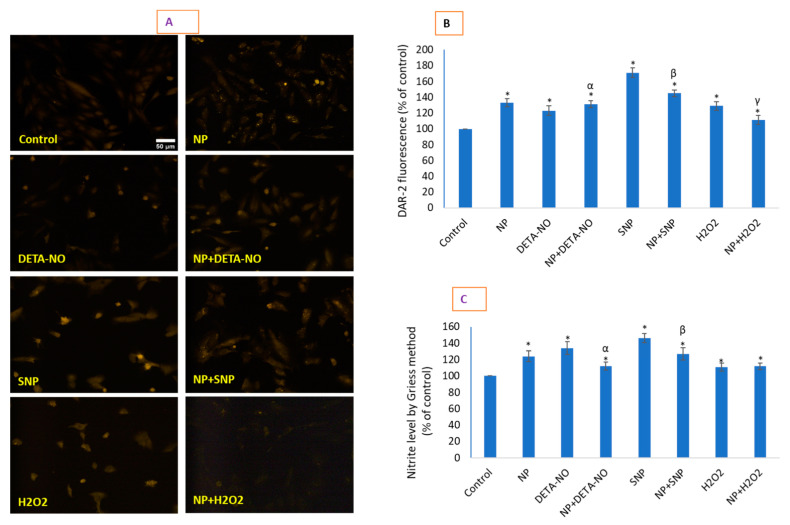
NO levels were measured by imaging live cells under incubation with NO-specific DAR-2 probe and indirectly by quantifying nitrite in culture media formed from NO using the Griess reagent. DAR-2 images and the fluorescence quantification are presented in (**A**,**B**), whereas measurement with the Griess reagent is given in (**C**). The scale bar, marked only in control images, represents 50 µm captured by a 20× objective. Data represented are means ± SD of three identical experiments (*n* = 3) made in triplicate. * Statistically significant difference as compared to the controls (*p* < 0.05). α, β, and γ, if present, denote the significant NP-mediated modulation of NO induced by DETA-NO, SNP, and H_2_O_2_, respectively (*p* < 0.05).

**Figure 6 molecules-26-05416-f006:**
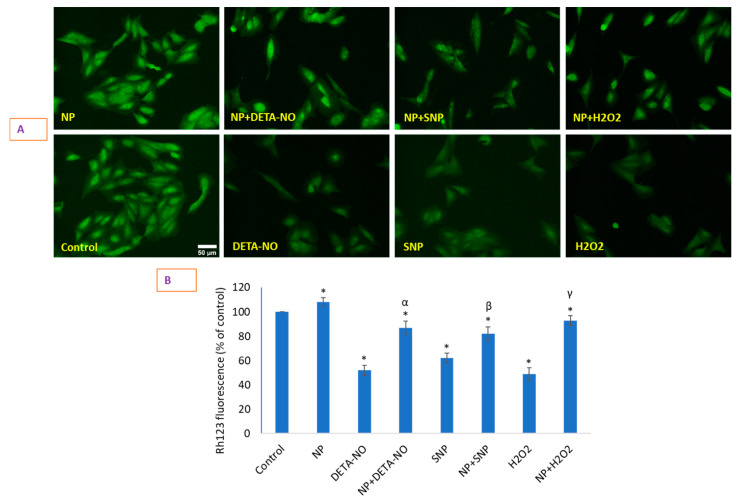
Potential gain or loss in MMP was determined by a Rh123 probe that emitted an increase or decrease in Rh123 fluorescence, respectively. Rh123 images and fluorescence quantification are presented in (**A**,**B**). The scale bar, marked only in control images, represents 50 µm captured by a 20× objective. Data represented are means ± SD of three identical experiments (*n* = 3) made in triplicate. * Statistically significant difference as compared to controls (*p* < 0.05). α, β, and γ, if present, denotes significant NP-mediated recovery of MMP that could be lost by DETA-NO, SNP, and H_2_O_2_, respectively (*p* < 0.05).

**Figure 7 molecules-26-05416-f007:**
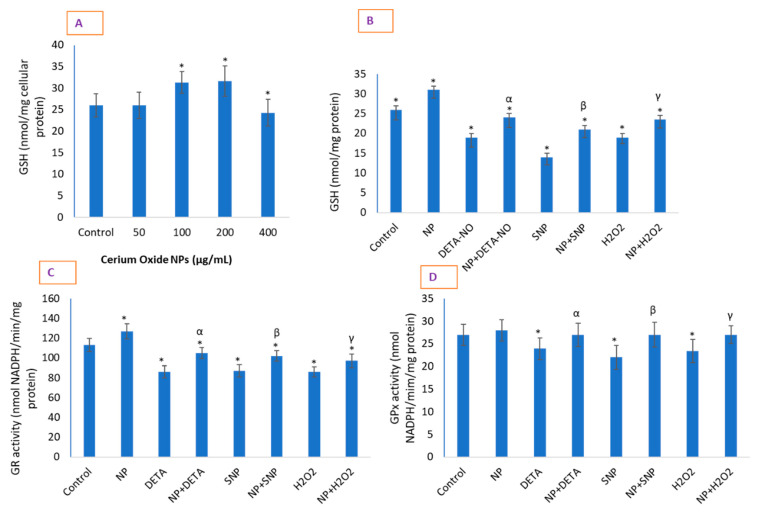
Concentration-dependent effect of CeO_2_ NPs on intracellular GSH (**A**). (**B**) Significant restoration of GSH that was otherwise significantly depleted by the IC50s of the two NO donors. (**C**,**D**) Modulatory behavior of CeO_2_ NPs on the activities of GR and GPx, respectively. Data are represented as means ± SD of three identical experiments (*n* = 3) performed in triplicate. * Statistically significant difference as compared to controls (*p* < 0.05). α, β and γ, if present, denote a significant NP-mediated replenishment of GSH that might have depleted by DETA-NO, SNP and H_2_O_2_, respectively (*p* < 0.05).

## Data Availability

Not applicable.

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
