# Peer review of "Anti-Inflammatory CeO2 Nanoparticles Prevented Cytotoxicity Due to Exogenous Nitric Oxide Donors via Induction Rather Than Inhibition of Superoxide/Nitric Oxide in HUVE Cells"

_molecules, 2021, doi:10.3390/molecules26175416_

Round 1
Reviewer 1 Report
In its present form the manuscript merit publication.
Author Response
In its present form the manuscript merit publication.
- Thanks for recommending to publish our manuscript.
Reviewer 2 Report
Dear Authors.
You have done an excellent job of research, the results sections mixed with the discussion does not distract or confuse, it is read in an agile way and the main idea is extracted in a timely manner. The only objection to this section is in sections 2.4 and 2.4, where you use the concept transiently, because there is no experiment where you develop a temporal kinetics of the processes or molecules searched, if the investigated effects are found, but not in a transient way, I suggest eliminating this word (transiently). The results have good order and structure, and are well described. The methodology is well described, and of adequate length. The methodologies chosen to test your hypothesis are adequate and allow to see clearly the results found.
The only section that brings me doubts is the introduction, it reads well until the presentation of the hypothesis and objectives of the research. However, the following lines do little to clarify the ideas expressed above. I suggest moving some of these explanations of the experiments to be performed to the methodology section, or shortening the length of the introduction.
Author Response
Dear Authors.
You have done an excellent job of research, the results sections mixed with the discussion does not distract or confuse, it is read in an agile way and the main idea is extracted in a timely manner. The only objection to this section is in sections 2.4 and 2.4, where you use the concept transiently, because there is no experiment where you develop a temporal kinetics of the processes or molecules searched, if the investigated effects are found, but not in a transient way, I suggest eliminating this word (transiently). The results have good order and structure, and are well described. The methodology is well described, and of adequate length. The methodologies chosen to test your hypothesis are adequate and allow to see clearly the results found.
- Thanks for the encouraging lines. We have removed the ‘transiently’ word from the manuscript though we would like to bring the kind attention of the reviewer that we carried DCF and DHE measurements in HUVE cells exposed with CeO2 nanoparticles at 100 µg/mL in a time-dependent manner (kindly see supplementary figure S1). Thanks for the comment.
The only section that brings me doubts is the introduction, it reads well until the presentation of the hypothesis and objectives of the research. However, the following lines do little to clarify the ideas expressed above. I suggest moving some of these explanations of the experiments to be performed to the methodology section, or shortening the length of the introduction.
- We once again thank to the reviewer for giving us important insight about effective introduction. As rightly suggested by the reviewer, we have cut some lines from the ‘introduction’ and placed it as N.B. after ‘cell culture and treatments’ in the section of ‘materials and methods’. Kindly see the modifications made in the revised manuscript. Thanks for the comment.
Reviewer 3 Report
The research work entitled “Anti-inflammatory CeO2 nanoparticles prevented cytotoxicity due to exogenous nitric oxide donors via induction rather than inhibition of superoxide/nitric oxide in HUVE cells” presents the cytoprotective potential of CeO2 NPs against cytotoxic nitric oxide (NO) donors and H2O2. The CeO2NP-mediated induction of ROS and/or NO could result in the activation of pro-survival signaling, which could prime cells to cope with exogenously induced oxidative stress. These works are interesting, however, the proof-of-concept for the synthesized material, original novelty, innovative insights, presentation, proposed mechanism, conceptualization, and discussion show insufficient explanation and poor clarity of presentation. Specific comments that may improve the article quality are given below:
- The quality of the English presentation is ambiguous and absence of clarity; the whole manuscript needs a tremendous English revision.
- The arrangement and captions quality of the presented figures should be greatly improved.
- The authors described that most rare-earth NPs have been reported to have angiogenesis and inflammation activities but lack the corresponding example, please provide it.
- The size of CeO2 NPs in TEM images looks aggregated, how did the authors define the distributed size?
- Please assign the d-spacing of CeO2 NPs in HR-TEM images.
- The morphology of CeO2 NPs in SEM images couldn’t seem to be assigned rhomboid- or cubical-like structures.
- Why the size of CeO2 NPs has great differences in DLS and TEM analysis, please explain.
- There are several confusing statements in the text:
- At line 108, the unit of Zeta may not present as “nm”.
- I miss the full name of “CSE” at line 209, please fill up it.
- In line 357, the “fig 7A and B” should be corrected to “fig 6A and B”.
- Generally, the cell toxicity assay should be evaluated up to 72 h, please consider it.
- Please provide the passage and number of donor pool of HUVECs.
- Please mark the scale bar of all cell images.
- In Fig. 3B & D, the TNF-a and IL-1B expression decreased slightly (<20%) when treated with CeO2 NPs, the results seem not reduced significantly.
- Please explain the decreasing results of NP+DETA-NO or NP+SNP in Fig, 4D.
- In Fig 5, there are some questions below:
- Please provide the bright field cell images.
- The DHE fluorescence between all groups looks not significant, and the DHE fluorescence graphics in Fig. 5D confused me, please expound in detail.
- The DHE and DCF fluorescence Fig 5A & B should show a similar trend, I am confusing with this result.
- In line 355, the authors wrote “CeO2 NP treatment has no effect on MMP in comparison with control cells”, but there is “*” in Fig. 6. Please elucidate it.
- The STD in Fig. 6 looks unreasonable, please re-calculate it again.
- There is a lot of possible factors that caused MMP down-regulation, such as withdrawal of growth factors or loss of the extracellular glucose supply, it’s not necessary NP-mediated O2 Please further clarify it.
Author Response
The research work entitled “Anti-inflammatory CeO2 nanoparticles prevented cytotoxicity due to exogenous nitric oxide donors via induction rather than inhibition of superoxide/nitric oxide in HUVE cells” presents the cytoprotective potential of CeO2 NPs against cytotoxic nitric oxide (NO) donors and H2O2. The CeO2 NP-mediated induction of ROS and/or NO could result in the activation of pro-survival signaling, which could prime cells to cope with exogenously induced oxidative stress. These works are interesting, however, the proof-of-concept for the synthesized material, original novelty, innovative insights, presentation, proposed mechanism, conceptualization, and discussion show insufficient explanation and poor clarity of presentation. Specific comments that may improve the article quality are given below:
- The quality of the English presentation is ambiguous and absence of clarity; the whole manuscript needs a tremendous English revision.
- Dear learned reviewer, the English of manuscript has been revised by MDPI language editing services during the first revision of this manuscript. We thought this should be satisfactory for a report to be published in scholarly journal. However, if the language needs further improvement at few occasions, we would like to request you to point out some language errors in the manuscript and would be happy to incorporate as per your direction. If it is substantial then simply mark (without improvement suggestion) those sentences that need improvement as this marking will help us in communicating with the MDPI language services regarding quality of their English language service. Thanks for commenting.
- The arrangement and captions quality of the presented figures should be greatly improved.
- We have modified some headings in result section as suggested by you and another reviewer. Kindly see the revision. Thanks for the comment.
- The authors described that most rare-earth NPs have been reported to have angiogenesis and inflammation activities but lack the corresponding example, please provide it.
- We have categorically mentioned rare-earth NPs to have mixed effects regarding angiogenesis and inflammation. Kindly see the beginning part of introduction of this manuscript where we have put reasonable amount of literature on CeO2 NPs that suggest about its pro- and anti-angiogenic property. Thanks for the comment.
- The size of CeO2 NPs in TEM images looks aggregated, how did the authors define the distributed size?
- We agree with the learned reviewer about aggregation of particles. We think that images could have been improved by taking extra precaution during grid preparation for TEM and SEM imaging. We would like to point out that TEM and SEM imaging are conducted by technical expert of electron microscope. We have to depend on them for TEM, SEM and other characterization data about nanoparticles. However, we have replaced previous SEM image with another SEM of CeO2 nanoparticles. We hope this should serve the purpose of deciphering shape of nanoparticles. Thanks for the comment.
- Please assign the d-spacing of CeO2 NPs in HR-TEM images.
- We have never ever used d-spacing like criterion so far in our previous publications since we have never encountered any correlation discussed in scientific literature between d-spacing and biological responses. Anyway, we discussed about d-spacing with our technician and made a request to provide information about this in light of reviewer comment. Kindly see it in the revision. Thanks.
- The morphology of CeO2 NPs in SEM images couldn’t seem to be assigned rhomboid- or cubical-like structures.
- Upon a careful observation of TEM and SEM image together, we were told by the technician that the overall shape of CeO2 nanoparticles occur as rhomboid- or cubical. In previous SEM image provided the cores and edges of nanoparticles are slightly less clear which have been replaced by another SEM image that looks clearer. We hope this should serve the purpose of deciphering shape of nanoparticles. Thanks for commenting.
- Why the size of CeO2 NPs has great differences in DLS and TEM analysis, please explain.
- To the best of our knowledge size analyzed by DLS and TEM can differ because DLS measures hydrodynamic sizes of nanoparticles that could be aggregated or clumped due to hydrophobic effect in aqueous media whereas TEM measure aero-dynamic sizes in dried condition. Thanks for the comment.
- There are several confusing statements in the text:
- At line 108, the unit of Zeta may not present as “nm”.
- Thanks for pointing out this typo-error. We regret for the any inconvenience happened to the reviewer. Kindly look the corrections made. Thanks once again.
- I miss the full name of “CSE” at line 209, please fill up it.
- CSE stands for ‘cigarette smoke extract’. We apologize for the inconvenience. We have used its full form in the revision. Thank for the comment.
- In line 357, the “fig 7A and B” should be corrected to “fig 6A and B”.
- Thanks to the reviewer’s sharp notice on this typo-error. Kindly see the correction. Thank for the comment.
- Generally, the cell toxicity assay should be evaluated up to 72 h, please consider it.
- We agree with the learned reviewer. Our study presented here clearly suggests that CeO2 nanoparticles are not toxic at mentioned concentrations when exposed for 24 h (and 48 h, data not shown). Thanks for the comment.
- Please provide the passage and number of donor pool of HUVECs.
- A passage number below 5 was used. HUVE cells were obtained from ATCC, US. Thanks for the comment.
- Please mark the scale bar of all cell images.
- We have used scale-bar only in control cells a convention found in countless published articles. Use of scale-bar in all cell images will be a tedious task with no potential additional benefit. Thanks for the comment.
- In Fig. 3B & D, the TNF-a and IL-1B expression decreased slightly (<20%) when treated with CeO2 NPs, the results seem not reduced significantly.
- We took a value of p < 0.05 as a significant difference in statistics. It means according to this criterion any difference of 5% or above between two compared groups is significant. Thanks for the comment.
- Please explain the decreasing results of NP+DETA-NO or NP+SNP in Fig, 4D.
- CeO2 NPs generated O2•− but at non-cytotoxic concentrations. That is O2•− inducing NPs did not cause reduction in cell viability in HUVE cells. However, O2•− generating CeO2 NPs caused a reduction of O2•− production due to the IC50s of the two NO donors and H2O2, thus suggesting a possible mechanism and implications for the NP-mediated production of O2•− (see lines 249-252 in the manuscript). This finding has been discussed with similar findings published in other articles (lines 252-263 in the manuscript).
We concluded that O2•− production, in conjunction with NO, could increase cells’ antioxidant status from the basal level increasing tolerance against toxicity of NO-donors. This promoted antioxidant capacity, as evidenced by an increase in GSH levels in NP-treated cells, is involved in allowing cells to tolerate the induced toxicity that might be due to exogenous oxidants and other toxic insults (lines 580-584 in the manuscript). Thanks for the comment.
Although, we have concluded this in the end of this report, we have added some lines in the results and discussion too to make it clearer. Kindly see the revision. Thanks for the comment.
- In Fig 5, there are some questions below:
- Please provide the bright field cell images.
- We have provided phase-contrast cellular images in fig 2 representing every treatment condition. Kindly see fig 2. Thanks for the comment.
- The DHE fluorescence between all groups looks not significant, and the DHE fluorescence graphics in Fig. 5D confused me, please expound in detail.
- We are happy to clarify that looking at figures with eye may not always give clear picture about differences, especially, in fluorescence. Our eye is not as sensitive as the light detecting devices. That’s why well established softwires are recommended to quantify data. Anyway, zoom-in images are clearer to eyes where one can easily observe fluorescence differences in nuclear and cytoplasmic areas of controlled and treated cells. Thanks for the comment.
- The DHE and DCF fluorescence Fig 5A & B should show a similar trend, I am confusing with this result.
- DHE measure specifically superoxide and DCF fluorescence measure ROS that may include many species of ROS including superoxide, H2O2 Again, cells may produce various ROS differently depending on cell types and interacting environment. Thanks for the comment.
- In line 355, the authors wrote “CeO2 NP treatment has no effect on MMP in comparison with control cells”, but there is “*” in Fig. 6. Please elucidate it.
- We thank to the reviewer’s sharp notice. By writing no effect we wanted to say no harmful effect. The significant sign “*” in Fig. 6 denotes a gain in MMP due to exposure of CeO2 nanoparticles relative to control HUVE cells. We have extended the phrase in the manuscript accordingly. Thanks once again.
- The STD in Fig. 6 looks unreasonable, please re-calculate it again.
- As suggested by reviewer, we recalculated std of fig 6. Thanks for commenting.
- There is a lot of possible factors that caused MMP down-regulation, such as withdrawal of growth factors or loss of the extracellular glucose supply, it’s not necessary NP-mediated O2 Please further clarify it.
- We fully agree with the learned reviewer. We did not differentially add or withdraw growth factors during HUVE cell culture or treatment except for the stated treatments. We never cut the supply of glucose during HUVE cell culture or treatment. Therefore, the difference was only due to treatment/exposure conditions mentioned and data, thus, obtained has been presented with respect to control of each experiment. Thanks for the comment.
Round 2
Reviewer 3 Report
Authors did not respond to all comments appropriately. Please provide answers/comments to the following questions.
- The quality of the English presentation is ambiguous and absence of clarity; the whole manuscript needs a tremendous English revision.
- The arrangement and captions quality of the presented figures should be greatly improved.
- The size of CeO2 NPs in TEM images looks aggregated, how did the authors define the distributed size? The size of CeO2 NPs in DLS and TEM analysis has great differences, please explain it.
- Please cite the literature on the d-spacing of CeO2 NPs in HR-TEM images.
- The morphology of CeO2 NPs in SEM images couldn’t seem to be assigned rhomboid- or cubical-like structures.
- Generally, the cell toxicity assay should be evaluated up to 72 h, please consider it.
- Please provide the passage and number of donor pool of HUVECs.
- In Fig. 3B & D, the TNF-a and IL-1B expression decreased slightly (<20%) when treated with CeO2 NPs, the results seem not reduced significantly, please elucidate it.
- Please explain the decreasing results of NP+DETA-NO or NP+SNP in Fig, 4D.
- In Fig 5, there are some questions below:
- Please provide the bright field cell images.
- The DHE fluorescence between all groups looks not significant, and the DHE fluorescence graphics in Fig. 5D confused me, please expound in detail.
- The DHE and DCF fluorescence Fig 5A & B should show a similar trend, I am confusing with this result.
- The STD in Fig. 6 looks unreasonable, please re-calculate it again.
- There is a lot of possible factors that caused MMP down-regulation, such as withdrawal of growth factors or loss of the extracellular glucose supply, it’s not necessary NP-mediated O2 Please further clarify it.
Author Response
Reviewer #3
- The quality of the English presentation is ambiguous and absence of clarity; the whole manuscript needs a tremendous English revision.
- We have told you during previous revision that English of the current manuscript had been already edited by MDPI Language editing services. In spite of this editing, we asked you to point out some errors from the manuscript text as a reference so that we can contact the Language editing services on that reference basis and request them review the language of the manuscript once again. But you have not pointed any line from the manuscript that needs “a tremendous English revision” according to you.
Kindly provide a couple of errors as example or accept that you are misusing the capacity of being reviewer.
- The arrangement and captions quality of the presented figures should be greatly improved.
- You have again put out a comment that we responded previously. Kindly suggest a possible change in any of “The arrangement and captions quality of the presented figures should be greatly improved” or accept that you are misusing the capacity of being reviewer.
- The size of CeO2 NPs in TEM images looks aggregated, how did the authors define the distributed size? The size of CeO2 NPs in DLS and TEM analysis has great differences, please explain it.
- We replaced the SEM image that looked aggregated to you during the revision. Now TEM image “looks” aggregated to you. Why did not you point this too during the first revision.
And where have you read that hydrodynamic and aerodynamic size of nanoparticles cannot or should not differ? It will be interesting to listen from you on this “discrepancy”. This becomes really a bizarre kind of comment if raised repeatedly even after a humble clarification during previous review.
- Please cite the literature on the d-spacing of CeO2 NPs in HR-TEM images.
- Can you tell us what do you understand by d-spacing and why is it so important in this manuscript? We would be grateful if you could enlighten us about the correlation of d-spacing of nanoparticles with the potential bio-response of nanoparticles. Looking to hearing from you.
- The morphology of CeO2 NPs in SEM images couldn’t seem to be assigned rhomboid- or cubical-like structures.
- It appears that you did not bother to review the revised version of our manuscript. In both TEM and SEM images, shape of nanoparticles can be easily seen. Anyway, we are curious about what shape do you see?
- Generally, the cell toxicity assay should be evaluated up to 72 h, please consider it.
- This is another comment that you raised in previous review. We very humbly responded you that this manuscript is not about toxicity of CeO2 nanoparticles but rather a potential protection from toxicity induced by other exogenous agents.
And for your limited information, there are countless publication on toxicity of numerous agents that have been carried out for a 24 h exposure duration only.
Therefore, we strongly believe that this comment is for the sake of commenting. It might had a value to improve the manuscript if raised with specific and logical reasons.
- Please provide the passage and number of donor pool of HUVECs.
- We have already explained this comment raised during previous review as “A passage number below 5 was used. HUVE cells were obtained from ATCC, US. Thanks for the comment”. It appears you did not even bother to read our responses and you just copy pasted your previous comments again. Kindly be responsible before wasting time of investigators.
- In Fig. 3B & D, the TNF-a and IL-1B expression decreased slightly (<20%) when treated with CeO2 NPs, the results seem not reduced significantly, please elucidate it.
- We have already explained this comment raised during previous review. In the text it reads as “NPs caused an inhibition of IL-1β by 12% (fig 3B) and TNF-α by 21% (fig 3C).” It is clear either you have not read our responses or not knowledge about basic statistics. In either case this is not our fault.
- Please explain the decreasing results of NP+DETA-NO or NP+SNP in Fig, 4D.
- We have already explained this comment raised during previous review. If you are not satisfied with the given explanation you could have better given your reasons.
- In Fig 5, there are some questions below:
- Please provide the bright field cell images.
- The DHE fluorescence between all groups looks not significant, and the DHE fluorescence graphics in Fig. 5D confused me, please expound in detail.
- The DHE and DCF fluorescence Fig 5A & B should show a similar trend, I am confusing with this result.
- The STD in Fig. 6 looks unreasonable, please re-calculate it again.
- There is a lot of possible factors that caused MMP down-regulation, such as withdrawal of growth factors or loss of the extracellular glucose supply, it’s not necessary NP-mediated O2 Please further clarify it.
- For comment 10 to 12, We have a common answer that we already responded to every comment. If you are not satisfied with the given explanation you could have better given your reasons. For better commenting, you are suggested to read our previous responses this time.
We would be delighted to entertain the reviewer again. Looking forward to reviewer constructive comments. As corresponding author, I would humbly suggest reviewer to read the responses this time if reviewer wants really be constructive. Thanks.
Round 3
Reviewer 3 Report
Thanks for replying most questions. However, here are some additional comments for this manuscript.
1 The presented figures should be greatly improved. Please make the bar graph to more formal graph and consistent format.
2 In terms of the TEM and DLS explanation, the authors proposed the NPs could be aggregate under aqueous media. How was the size changed when CeO2 NPs is in the presence of culture media, does it affect the cell uptake efficiency?
3 The d-spacing value for the newly synthesized nanomaterials is quite important evidence for materials identification, please provide it.
4 Please provide a convincing rationale for the 24h cell toxicity assay, why not shorter or longer?
Author Response
- The presented figures should be greatly improved. Please make the bar graph to more formal graph and consistent format.
- We are really shocked of this comment. Are not the figures provided in the manuscript clear to a reader? Learned reviewer should specifically guide us on how are the bar graphs informal and inconsistent and what should be done to make it formal and consistent. This is really a point where we are clueless.
- In terms of the TEM and DLS explanation, the authors proposed the NPs could be aggregate under aqueous media. How was the size changed when CeO2 NPs is in the presence of culture media, does it affect the cell uptake efficiency?
- DLS measurement is not a direct evidence like images provided by microscopes (i.e. camera) to which one can “see”. To the best of our experience and knowledge, most nanoparticles (or any particle) are aggregated in aqueous media and have different hydrodynamic sizes than the size observed in aerodynamic form by microscopy. Even hydrodynamic size of same nanoparticles may (or may not) differ in one aqueous media like culture media without serum from the hydrodynamic size measured in culture media with We can conclude that some components present in aqueous media can have dramatic effect on the cumulative non-covalent interaction occurring among nanoparticles and hydrophobicity experienced by particles when present in (polar) aqueous solvent. Degree of aggregation of nanoparticle will be further affected by nanoparticle’s own composition and surface charge.
Any expert in the field of “nanotoxicology and nanoscience” are supposed to have basic idea behind different measurements of hydrodynamic and aerodynamic size of nanoparticles, and, also the different degree of possible aggregation in different aqueous media. We did not assay cellular uptake of nanoparticle as we lack a dependable facility currently.
- The d-spacing value for the newly synthesized nanomaterials is quite important evidence for materials identification, please provide it.
- Now the learned reviewer itself agree that d-spacing value is quite important evidence for materials identification. We would agree putting the d-spacing value and even other material parameter for reporting a novel material. The material we are reporting is the nanoparticle of CeO2 which has been synthesized by more than a single method for a decade and should not be regarded a novel material now.
We have had, however, provided d-spacing value during previous revision just due to insistence of reviewer despite knowing that d-spacing has nothing to discuss in the outcome of bio-response if once a materials composition is characterized.
- Please provide a convincing rationale for the 24h cell toxicity assay, why not shorter or longer?
- We reassert that our study is not about potential toxicity of CeO2 nanoparticles rather its protective potential was evaluated against exogenous toxicants for a 24 h duration. Exogenous toxicants included in this study belonged to different category such as inflammation-inducing lipopolysaccharide (LPS) and cytotoxicity-inducing nitric oxide donor and oxidant H2O2. We have done nothing un-usual in terms of exposure period.
Investigators may choose any time duration of in vitro exposure (i.e. from a few hours to few days) for a particular agents/compounds depending on the nature (or intensity) of bio-response or the nature of a particular parameter under consideration. These are very basic facts in investigative biology the description of which is beyond space permitted here.
This manuscript is a resubmission of an earlier submission. The following is a list of the peer review reports and author responses from that submission.
Round 1
Reviewer 1 Report
The manuscript is very interesting and is well-structured in general. The Authors conducted a number of assays to reveal that anti-inflammatory effect of CeO2 nanoparticles prevent cytotoxicity against exogenous nitric oxide donors via induction, rather than inhibition, of superoxide/nitric oxide in HUVE cells. In my opinion it could be interesting for a reasonable number of scientists. However as it was ponited by the Authors further studies are needed to present the mechanism of biological response of the new synthesised NP.
Minor:
According to the figures 2,5 and the y-axis should be corrected to be readable.
Furher experiments that are needed to show the mechanism of new synthesised NP action should be discussed.
In my opinion, after this correction, the manuscript merit publication.
Reviewer 2 Report
In my opinion the manuscript needs to be extensively revised by a fluent english speaker in close collaboration with the authors before even be considered for review. In its current form the level of english is very poor and makes the paper nigh on impossible to follow.
Furthermore, there are some serious flaws also in the very basic scientific nomenclature of the paper: the superoxide anion is written throughout as O2-. and it is unclear what the authors mean for O2- . Some of the figures are overlapped with the legends and some graphs have the axes overlapped with the text.
In summary, in my opinion the paper is not suitable to be considered for publication on Molecules.
Reviewer 3 Report
There is an excellent experimental study in wich the authors aim to investigate the level of antiinflammatory nanoparticles CeO2 cytoprotective action against the exogenous oxidative stress with introduction that outlines concisely and appropriately the pathway investigated,clear results ,well structured discussion with the appropriate references and with excellent in quality figures and necessary, informative and clear tables.
Anti-inflammatory CeO2 nanoparticles prevented cytotoxicity 2 against exogenous nitric oxide donors via induction, rather 3 than inhibition, of superoxide/nitric oxide in HUVE cells